# Characterization of Fish Assemblages and Standard Length Distributions among Different Sampling Gears Using an Artificial Neural Network

Tae-Sik Yu [1], Chang Woo Ji [1], Young-Seuk Park [2], Kyeong-Ho Han [3] and Ihn-Sil Kwak [1,4,*]

[1] Fisheries Science Institute, Chonnam National University, Yeosu 59626, Korea
[2] Department of Biology, Kyunghee University, Seoul 02447, Korea
[3] Department of Aquaculture, Chonnam National University, Yeosu 59626, Korea
[4] Department of Ocean Integrated Science, Biology, Chonnam National University, Yeosu 59626, Korea
* Correspondence: iskwak@chonnam.ac.kr

**Abstract:** Several sampling gears are used to collect fish in the lentic ecosystem. The collected fish differ in their characteristics and community structure depending on the sampling gear. The objectives of this study were to 1) compare the community structure of fish assemblages sampled using four sampling gears (kick net, cast net, gill net, and fyke net) in the Singal (SG), Yedang (YD), and Juam (JA) reservoirs, and 2) to understand the characteristics of fishes collected by each sampling gear. A total of 1887 individuals of 14 species, 9113 individuals of 15 species, and 9294 individuals of 27 species were collected, respectively, from the SG, YD, and JA reservoirs. Among the four sampling gears tested, the fyke net collected the largest numbers of species and individuals, while the gill net collections had the highest diversity index. The results obtained with the self-organizing map (SOM) provided a more detailed characterization of the sampled fish than the metrics that are typically used to evaluate sampling gears. In particular, SOM analysis showed a similar pattern of the standard length of fish and sampling gear. Since each sampling gear has unique characteristics, the selection of an appropriate sampling gear should be based on the study objectives and features of the sampling sites.

**Keywords:** fish assemblage; lentic ecosystem; sampling gears; fishing gear; SOM



## 1. Introduction

Fish are the top consumers in aquatic ecosystems and are adapted to various habitats. Fish assemblages are widely studied to understand and interpret changes in freshwater ecosystems, including changes in water quality and structural habitat quality. The sampling and identification of fish species is a fundamental step in studying aquatic ecology and has significant applications in fisheries' resource assessments. Information obtained through sampling fish communities is essential for environmental assessments as well as for the conservation of biodiversity and the efficient management of aquatic resources [1–3].

Various sampling gears have been developed for collecting fish [4], and each sampling gear has specific characteristics. The traditional sampling gears most commonly used in lentic ecosystems include the trap net, trammel net, gill net, seine net, and fyke net [5]. Sampling efficiency varies depending on sampling gear, fish species, fish size, and habitat [6]. The relative abundance and species diversity of collected fish may not accurately reflect the proportions of fishes in natural assemblages [7]. Appropriate fishing gear should be selected, since sampling gear characteristics determine the number of species, the number of individuals, and the community composition of the catch. Multiple fish sampling gears are required in lentic habitats because the lentic ecosystem has distinct physicochemical zones [2]. The lentic ecosystem is divided into four zones based on water depth: the littoral zone, the limnetic zone, the profundal zone, and the benthic zone. Researchers should

select appropriate sampling gears for each zone or target fish species. For instance, cast and kick nets have been widely used in the littoral zone, while gill and fyke nets have been used in the limnetic and profundal zones [6].

Numerous studies evaluating the efficiency of these sampling devices in lentic ecosystems have been conducted [6,8–11]. Jackson and Harvey [9] compared the relative abundance and patterns of covariation of fish among minnow traps, plastic traps, fine- and coarse-mesh trap nets, and multi-mesh gill nets in 43 lakes. However, they found conflicting results for relative species abundance among the sampling gears. McInerny and Cross [10] showed differences in relative species abundance between day and night sampling gears, and Han et al. [6] found differences among sampling gears in terms of their relative efficiency and the fish species collected. Mueller et al. [11] compared nine sampling methods in lentic flood-plain habitats with respect to habitat characteristics, habitat type, and sampling duration. Previous studies suggested differences in the fish species collected according to the sampling gear, but we tried to present the characteristics (e.g., standard length) of the fish collected according to the sampling gear.

The self-organizing map (SOM) is an artificial neural network technique that relates similar features of high-dimensional datasets to a reduced-dimensional set of output responses [12–15]. SOM can visualize a large amount of cluster data [15] and recognize patterns of fish characteristics [16,17]. The main advantage of SOM is that researchers can easily analyze the map to determine the data's structure [18]. SOM is widely used in ecology because it is a straightforward technique for visualizing and displaying large amounts of cluster data. Several studies have used SOM to examine the relationship between sampling gears and fish communities [19–21].

This study aimed to identify the characteristics of four sampling gears and the relationship between the sampling gears and fish species. We first explored fish assemblages and their standard length distributions in lentic ecosystems based on four different sampling gears, while the SOM was subsequently used to classify the sampling traits of dominant fish species.

## 2. Materials and Methods

### 2.1. Study Location

To analyze the characteristics of the fish collected by the four sampling gears, three reservoirs were selected based on the lake area, as defined in the Biomonitoring Survey and Assessment Manual [22]. Sampling was conducted at one site in the Singal reservoir (SG), which has an area of approximately 2.58 km$^2$ and belongs to the small-sized (<3 km$^2$) reservoir category (Figure 1a). Sampling was conducted at three sites (inflow, middle-flow, and outflow regions) in the Yedang reservoir (YD), which has an area of approximately 9.9 km$^2$ and belongs to the medium-sized (3–50 km$^2$) reservoir category. Sampling was also conducted at three sites (inflow, middle-flow, and outflow regions) in the Juam reservoir (JA), which has an area of about 1010 km$^2$ and belongs to the large-sized (>50 km$^2$) reservoir category. Nonpoint pollution sources have increased in the SG and YD due to urban and industrial development in the SG and agricultural runoff in the YD, while water quality is managed and maintained in the JA, which is categorized as a water source conservation area.

To evaluate the water quality at each site, we measured the surface water (<50 cm depth) temperature, dissolved oxygen (DO, mg/L), potential of hydrogen (pH), and conductivity (EC) using a hand-held multiparameter meter (YSI, Professional Plus, Ohio, USA). Total organic carbon (TOC) concentrations were measured by the high-temperature (850 °C) combustion catalytic-oxidation method using a TOC analyzer (vario TOC cube, Langenselbold, Germany). For total nitrogen (T-N), total phosphorus (T-P), and Chl-a concentration measurements, water samples were filtered through a 0.45 μm pore-size membrane (Advantec MFS membrane filter, Dublin, OH, USA) and measured using a UV spectrophotometer. The suspended solids (SS) were filtered through a GF/C filter according to Park and Jung [23].

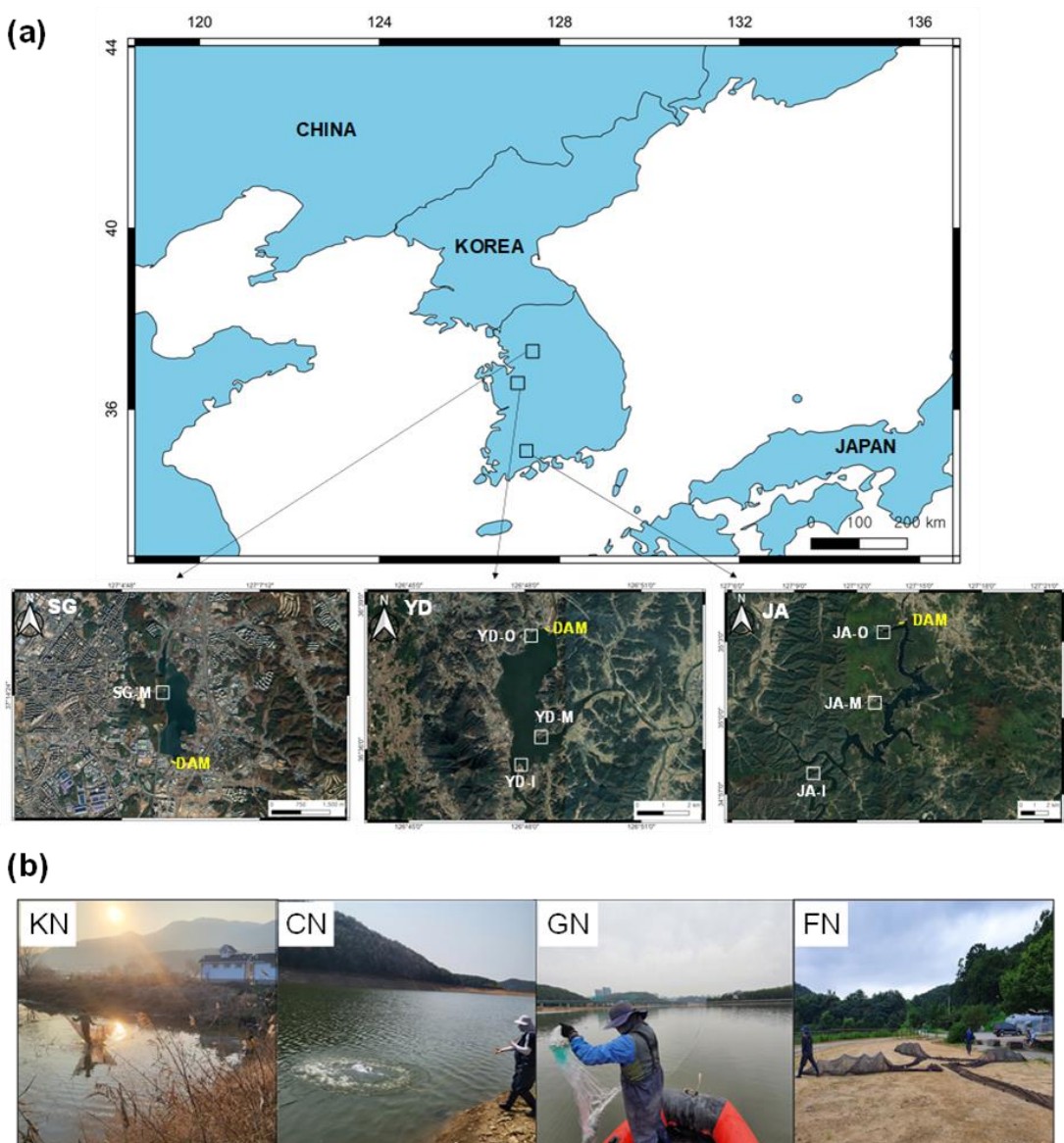

**Figure 1.** (**a**) A map of the Singal (SG), Yedang (YD), and Juam (JA) reservoirs with the seven littoral sampling sites; (**b**) The pictures of four sampling gears for sampling fishes from the SG, YD, and JA in Korea. I: Inflow region, M: Middle-flow region, O: Outflow region, KN: Kick net, CN: Cast net, GN: Gill net, FN: Fyke net.

### 2.2. Fish Collection and Identification

Fish were sampled from August 2020 to October 2021. Sampling was conducted five times in the SG, three times in the YD, and four times in the JA (Figure 1a). In the littoral zone (water depth ≤ 3 m), a cast net (6 × 6 mm, 10 times per site) and kick net (4 × 4 mm, 30 mins) were used in the daytime, and a gill net and fyke net were used overnight (Figure 1b). The gill net was 100 m in total length and 1.5 m in height, and the stretched mesh sizes were 45 mm and 12 mm. The fyke net frame was constructed of three pockets with 3 mm mesh size netting and had a 20 × 2.4 m high lead. The fish sampling was carried out using the same standard methods for quantitative investigation in three reservoirs. All sampled fish were identified to species [24], measured (total length (TL) and standard length (SL)), and weighed (Multi-use Balance, Daihan, Republic of Korea). The scientific names for all species followed FishBase [25] and Fish Species of Korea [26].

### 2.3. SOM and Data Analysis

Water quality was expressed as the mean ± deviation for all data, divided by each reservoir. Subsequently, a multiple comparison test (Tukey) was conducted to compare significant water quality differences among the SG, YD, and JA. The diversity (H') and dominance (D) indexes were calculated based on the results of the fish species identification. The diversity index was calculated according to Shannon information entropy [27], and the dominance index was calculated according to Simpson [28].

An SOM is an unsupervised artificial neural network composed of input and output layers connected by weight vectors [13]. In this study, we used SOM to categorize the characteristics of the fish collected by each sampling gear. Data on the sites, sampling period, standard length, biomass, and sampling gears (kick net, cast net, gill net, and fyke net) were natural log-transformed and used as input data for training the SOM. We used 70 (N = 7 × 10) SOM output units to show the most biologically relevant results. After the learning process of SOM was completed, the Bray–Curtis distance was calculated using weights and classified through hierarchical clustering using the Ward-linkage method. The statistical analyses were performed using R software (https://www.r-project.org/, accessed on 17 February 2022; version 4.0.5), using the packages "kohonen" [29] for SOM and "vegan" [30] for cluster analysis.

### 3. Results

#### 3.1. Water Quality

The mean of water temperature, pH, DO, T-P, SS, and Chl-a did not differ significantly among the three reservoirs ($p > 0.05$), whereas the mean EC, TOC, and T-N showed a significant difference ($p < 0.001$). The mean water temperature was highest in the JA at 23.1 ± 5.3 °C. The mean DO, EC, T-N, T-P, and SS values were highest in the SG. The mean pH, TOC, and Chl-a values were highest in the YD (Figure 2).

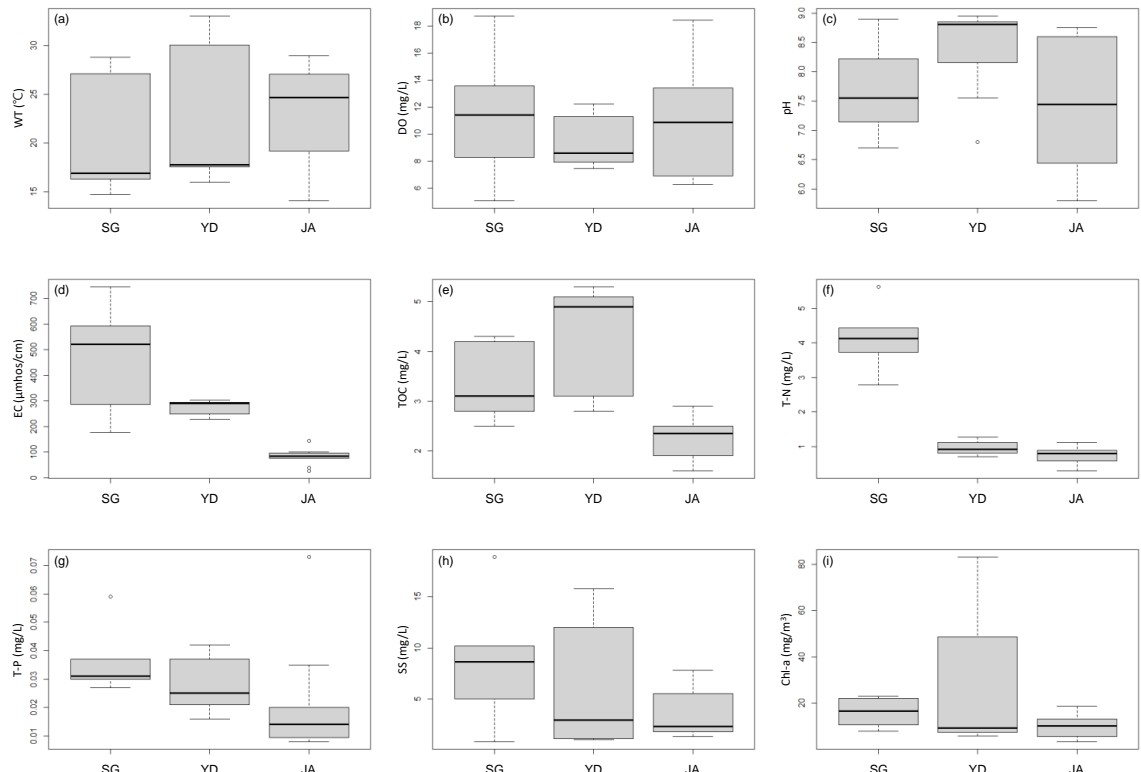

**Figure 2.** Mean values of water temperature (**a**), dissolved oxygen (**b**), potential of hydrogen (**c**), electrical conductivity (**d**), total organic carbon (**e**), total nitrogen (**f**), total phosphorus (**g**), suspended solids (**h**), and chlorophyll a (**i**) measured from the SG, YD, and JA in Korea.

*3.2. Fish Community Composition in Each Survey Reservoir*

The community structure, abundance, and diversity of fish in the three reservoirs were studied using the kick net, cast net, gill net, and fyke net (mean depth < 3 m). During the study, 31 species of fish and 20,294 individuals (337,177.0 g total weight) were recorded from the three reservoirs.

In the SG, which is categorized as a small-sized reservoir, 13 species of fish were collected (Table 1). A total of 1887 individual fish with a total biomass of 78,993.3 g were captured. The dominant species were *Pseudorasbora parva* (49.4%), *Micropterus salmoides* (20.0%), and *Carassius carassius* (9.5%). The standard lengths of the major species ranged from 23 to 123 mm (mean $51.4 \pm 24.7$ mm, *n* = 129) for *Lepomis macrohirus*, from 71 to 132 mm (mean $94.8 \pm 12.9$ mm, *n* = 338) for *M. salmoides*, and from 60 to 345 mm ($154.6 \pm 80.2$ mm, *n* = 171) for *C. carassius* (Figure 3a).

**Table 1.** The list and the number of fish individuals collected from the three reservoirs. These species were collected with a kick net, cast net, gill net, and fyke net.

| Fish Species | Reservoirs | | | Total | * R.A. (%) |
|---|---|---|---|---|---|
| | Singal | Yedang | Juam | | |
| *Cyprinus carpio* | 55 | 3 | 7 | 65 | 0.3 |
| *Carassius carassius* | 179 | 35 | 184 | 398 | 2.0 |
| *Carassius cuvieri* | 41 | 12 | 21 | 74 | 0.4 |
| *Channa argus* | 1 | | 1 | 2 | <0.1 |
| *Acheilognathus yamatsutae* | | | 6 | 6 | <0.1 |
| *Acanthorhodeus chankaensis* | | 1 | 106 | 107 | 0.5 |
| *Tanakia lanceolata* | | | 1 | 1 | <0.1 |
| *Hemibarbus labeo* | | | 55 | 55 | 0.3 |
| *Hemibarbus longirostris* | | 4 | | 4 | <0.1 |
| *Pseudogobio esocinus* | | 7 | 1 | 8 | <0.1 |
| *Pseudorasbora parva* | 933 | 48 | 15 | 996 | 4.9 |
| *Squalidus chankaensis tsuchigae* | | | 113 | 113 | 0.6 |
| *Squalidus japonicus coreanus* | 12 | | | 12 | 0.1 |
| *Pungtungia herzi* | | | 23 | 23 | 0.1 |
| *Microphysogobio yaluensis* | | | 27 | 27 | 0.1 |
| *Hemiculter leucisculus* | 7 | 464 | 596 | 1067 | 5.3 |
| *Zacco platypus* | 92 | 1 | 2400 | 2493 | 12.3 |
| *Opsariichthys uncirostris amurensis* | | | 121 | 121 | 0.6 |
| *Nipponocypris temminckii* | | | 2 | 2 | <0.1 |
| *Cobitis lutheri* | | 1 | | 1 | <0.1 |
| *Cobitis tetralineata* | | | 12 | 12 | 0.1 |
| *Misgurnus anguillicaudatus* | | 2 | | 2 | <0.1 |
| *Silurus asotus* | 2 | 4 | 9 | 15 | 0.1 |
| *Tachysurus fulvidraco* | | | 7 | 7 | <0.1 |
| *Hypomesus olidus* | | | 2421 | 2421 | 11.9 |
| *Odontobutis interrupta* | 1 | | 1 | 2 | <0.1 |
| *Rhinogobius brunneus* | 44 | 12 | 26 | 82 | 0.4 |
| *Rhinogobius giurinus* | | | 1 | 1 | <0.1 |
| *Siniperca scherzeri* | | | 1 | 1 | <0.1 |
| *Micropterus salmoides* | 377 | 109 | 97 | 583 | 2.9 |
| *Lepomis macrochirus* | 143 | 8410 | 3040 | 11,593 | 57.1 |
| Number of individuals | 1887 | 9113 | 9294 | 20,294 | |
| Number of species | 13 | 15 | 27 | 31 | |
| Biomass (g) | 78,993.3 | 82,013.8 | 176,169.9 | 337,177.0 | |

* R.A., Relative abundance.

In the YD, which is categorized as a medium-sized reservoir, 15 species of fish were collected. A total of 9113 individual fish with a total biomass of 82,013.8 g were captured (Table 1). The dominant species, *L. macrochirus*, accounted for 8410 (92.3%) of the total 9113 individuals captured. *Hemiculter eigenmanni* and *M. salmoides* were the next two most abundant species; however, they represented only 5.1% and 1.2%, respectively, of the total number of fish collected. The standard lengths of the major species ranged from 22 to 137 mm (mean $54.4 \pm 31.8$ mm, *n* = 1335) for *L. macrochirus*, from 42 to 315 mm (mean $97.4 \pm 57.3$ mm, *n* = 101) for *M. salmoides*, and from 160 to 260 mm ($224.0 \pm 23.5$ mm,

*n* = 33) for *C. carassius* (Figure 3a). The SL range of *C. carassius* collected in the YD was the narrowest among the three reservoirs.

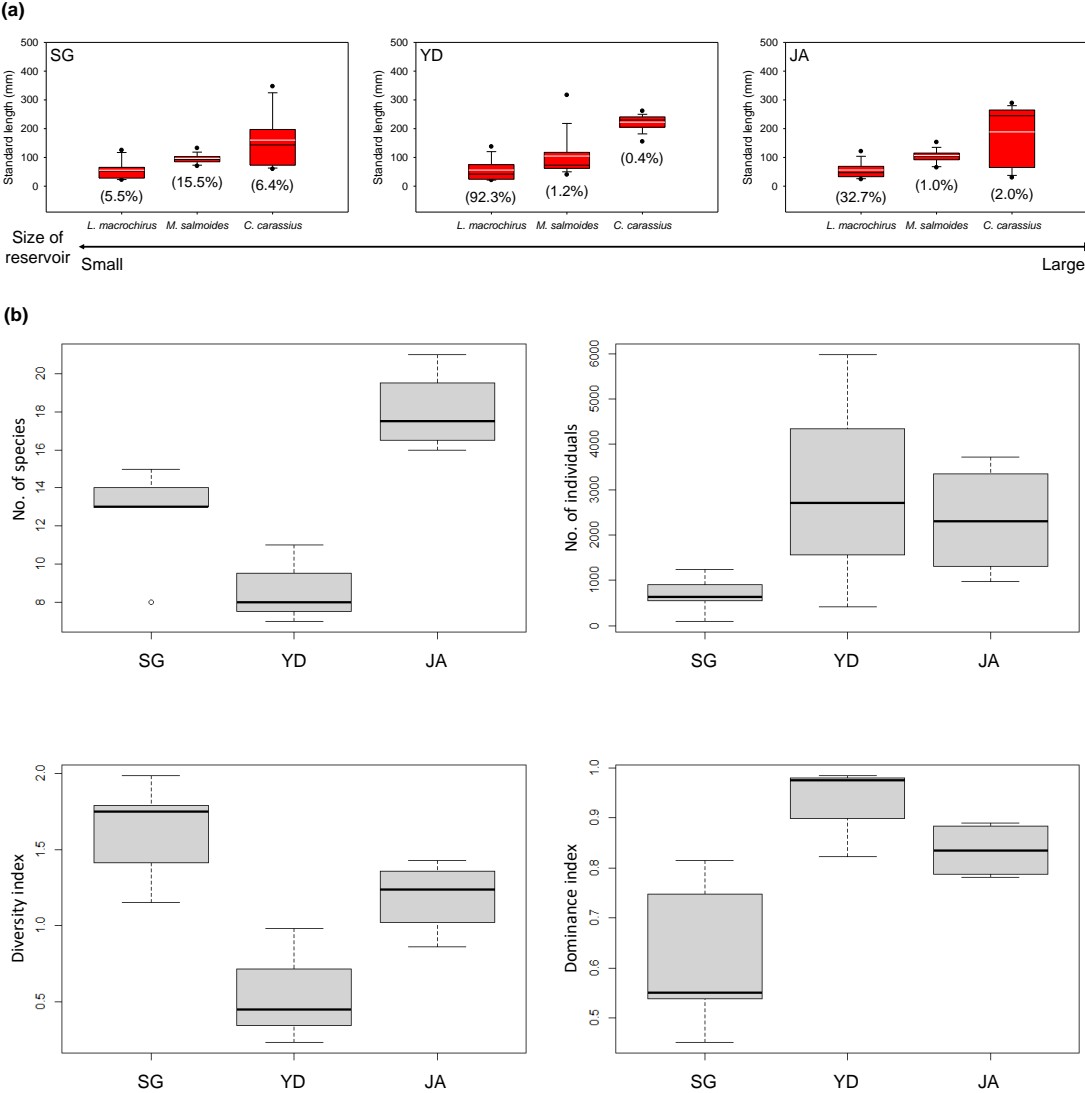

**Figure 3.** (**a**) The range of the standard length for the major species sampled from the SG, YD, and JA in Korea. The black lines represent the median value, and the white lines represent the average value. A number between brackets means the relative abundance. (**b**) The mean community structure values from the SG, YD, and JA in Korea.

The JA, which is categorized as a large-sized reservoir, had the highest abundance, biomass, and number of fish species (Table 1). The dominant species were *L. macrochirus*, *Hypomesus olidus*, and *Zacco platypus*. The standard lengths of the major species ranged from 25 to 117 mm (53.7 ± 22.9 mm, *n* = 1734) for *L. macrochirus*, from 65 to 152 mm (103.6 ± 18.9 mm, *n* = 89) for *M. salmoides*, and from 34 to 285 mm (192.3 ± 91.2 mm, *n* = 165) for *C. carassius* (Figure 3a).

The average number of species per number of sampling events was compared among the three reservoirs (Figure 3b). The mean number of fish species collected per number of samplings was highest in the JA (18 species) and was lowest in the YD (8.7 species). Among the three reservoirs, the YD had the highest mean number of individuals (3037.7) and the highest dominance index (D = 0.93). The mean diversity index was highest in the SG (H′ = 1.19) and was lowest in the YD (H′ = 0.56).

### 3.3. Extracting Fish Assemblage Using SOM

The results of the SOM analyses showed four clusters (structure of SOM map size: 7 × 10) of fish individuals based on sampling gears (Figure 4a). Each cluster was mainly associated with the characteristics of the sampling gears. For instance, cluster 1 mainly consisted of the fish sampled by the fyke net and kick net, cluster 2 consisted of those collected by the fyke net and cast net, cluster 3 consisted of those collected by the cast net and gill net, and cluster 4 consisted of those collected by the gill net. These distribution patterns show the characteristics of sampling gears. The fishes in the right-middle part of the SOM map were mainly large fish caught by the gill net, whereas the value of fish size decreased in the counterclockwise direction and consisted of fish caught by the kick net and fyke net.

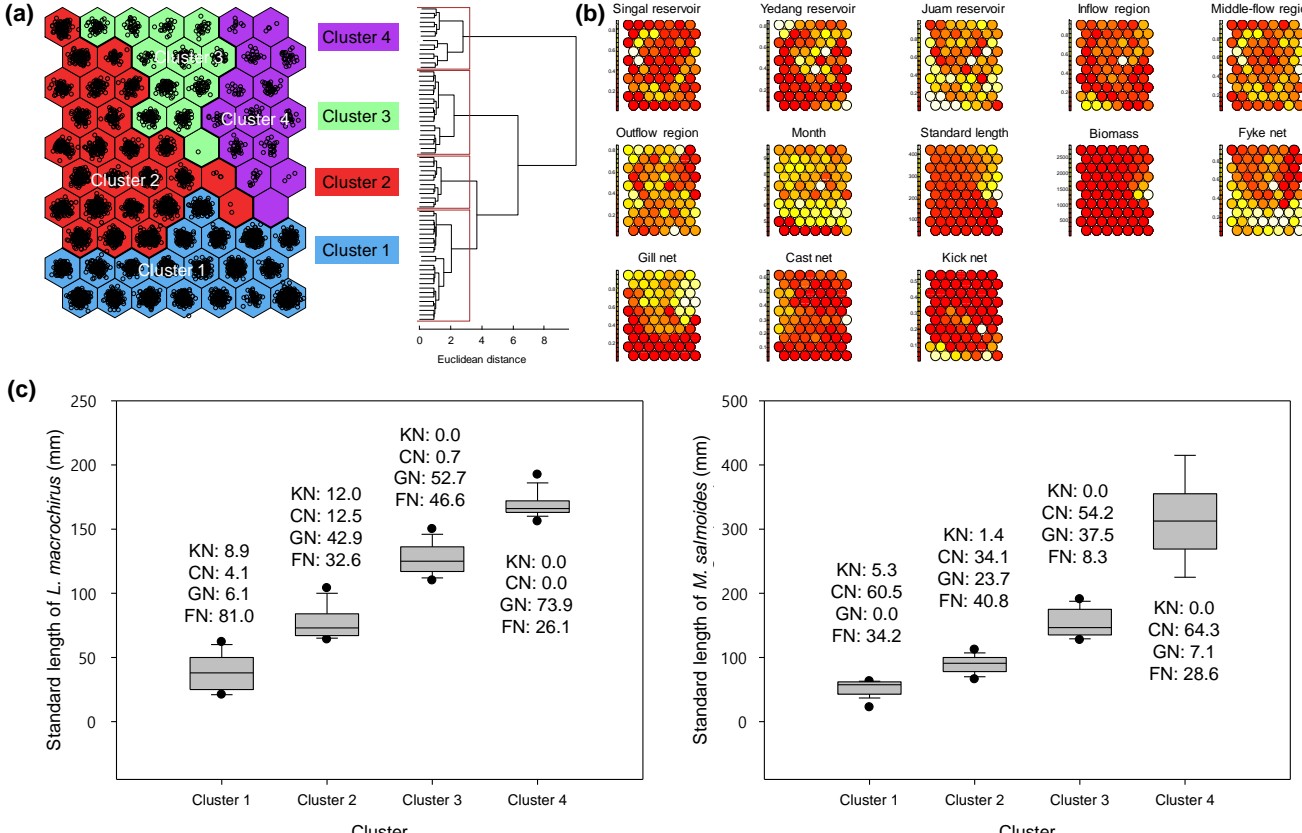

**Figure 4.** (**a**) Fishes sampled in the SG, YD, and JA were assigned to 70 self-organizing map (SOM) output neurons arranged into a two-dimensional grid (10 × 7). The neurons were grouped into four clusters with hierarchical cluster analysis and U-matrix values; (**b**) Component planes display the contribution of each variable to the classification of sampling sites, gears, and the standard length and biomass of fish. White neurons represent high values of each variable, whereas red neurons are for low values. The values were calculated during the learning process of the network; (**c**) A standard length distribution of dominant species (*Lepomis macrochirus* and *Micropterus salmoides*) by each cluster. The numbers indicate the percentage of fish caught by the sampling gear.

The contribution of each input variable (study site, standard length, biomass, and sampling gear) to the classification of fishes is shown on the SOM map (Figure 4b). Among the input variables, standard length displays a pattern similar to the gill net, but the fyke net shows the opposite pattern.

The distribution of the standard length for each cluster of *L. macrochirus* and *M. salmoides*, which were two of the dominant species in this study, is shown in Figure 4c. The standard length range of *L. macrochirus* was 22–59 mm (*n* = 1230, median = 38 mm) in cluster 1 and gradually increased to 65–103 mm (*n* = 629, median = 73 mm) in clus-

ter 2, 111–150 mm ($n$ = 261, median = 125 mm) in cluster 3, and 160–190 mm ($n$ = 21, median = 166 mm) in cluster 4. Similarly, the standard length range of *M. salmoides* gradually increased from cluster 1 to cluster 4. These distributions of standard length showed that each cluster on the SOM map was divided according to the standard length of the fish.

## 4. Discussion

In order to analyze the characteristics and patterns of fish collected by four different sampling gears (kick net, cast net, gill net, and fyke net), we used each type of gear to collect fish from the SG, YD, and JA in Korea. A total of 20,294 individuals representing 32 species and having a total weight of 337,177.0 g were collected from the three reservoirs. In the SG, 14 species, 1887 individuals, and 78,993.3 g of fish were collected; in the YD, 15 species, 9113 individuals, and 82,013.8 g of fish were collected; and in the JA, 27 species, 9294 individuals, and 176,169.9 g of fish were collected. Among the four sampling gears, the fyke net collected the highest number of species (24) and highest number of individuals (16,630), while the kick net collected the lowest number of species (12) and the lowest number of individuals (827).

The SG, YD, and JA are lentic ecosystems representing different reservoir's sizes and are impacted by different sources of pollution. The community structure of fish in each of the three reservoirs may have been influenced by the reservoir's size, sources of pollution, and water quality. Nonpoint source pollution due to stormwater runoff from urban developments and agricultural operations has increased the nitrogen and phosphorus levels in the aquatic ecosystem [31]. T-N and T-P, two of the water quality parameters measured in each reservoir, are important factors in controlling the nutrient structure and species composition of aquatic ecosystems [32–34]. High T-N and T-P values increase the populations of phytoplankton [35], a major food source for planktivorous fish [36]. The high T-N and T-P concentrations in the SG would have increased the production of phytoplankton, the primary food source for *P. parva* and other species of planktivorous fish in the lentic ecosystem, as reflected in the dominance of *P.parva* and the overall structure of the fish assemblage in the SG.

The introduction of nonnative fish affected the diversity and dominant species in each reservoir. In particular, *L. macrochirus* and *M. salmoides*, two of the dominant species in every reservoir, were nonnative fish introduced to Korea from the U.S.A. as food resources and for economic reasons [37,38]. However, both are considered to be ecosystem disturbance species due to their rapid increase in freshwater ecosystems and are known to affect species diversity negatively [39]. Consistent with previous studies that showed that dominance by *L. macrochirus* reduced species diversity, we confirmed that diversity was low (H' = 0.37) in the YD, where the relative abundance of *L. macrochirus* was 92.3%.

As tools for collecting fish that live mainly in shallow water habitats, the kick net and cast net are the most commonly used sampling gears to study fish communities in freshwater ecosystems [40,41]. The kick net and cast net are affected by the water depth of the sampling area, the direction of fish movement, and the sinking speed of the sampling gear [42]. It can increase species diversity by collecting fish that inhabit the littoral zone or benthic zone. Gill nets have been widely used as a research tool to sample fish populations [43]. The fishes caught in gill nets are medium-sized or larger, and the shape of the fish, the presence or absence of spines on the dorsal fin, the body depth, and the girth of the fish significantly influence the catch [44,45]. Gill nets tend to underestimate the abundance of species that have few external projections or rigid structures and those that have more sedentary habits, as well as undersized individuals [46]. *Rhinogobius brunneus*, *Cobitis tetralineata*, and *H. olidus* were collected from the kick net, cast net, and fyke net but not from the gill net, illustrating the species and size selectivity of the gill net. The fyke net is a passive sampling device that is effective at collecting large numbers of mobile fish from the littoral zone [8]. Due to its ability to capture large numbers of species and individuals, the fyke net is commonly used for fishery assessments [47]. Among the fish collected in the fyke net, 10,270 individuals of *L. macrochirus* were collected, indicating a

relative abundance of 60.3%. In addition, the dominance of the fish sampled in the fyke net was 0.75, reflecting the fyke net's high species selectivity.

A self-organizing map (SOM) was used to visualize patterns among the sampled fish and characterize them based on sampling gear variables. The network was classified into four clusters based on the sampled fishes in the SOM layer. We then predicted correlations between the four clusters of fish and the variables in the output layer of the network. Several studies have examined the relationship between fish sampling gears and assessments of the sampled fish community [48–52]. These studies have shown that various sampling gears strongly affect the assessments of fish assemblages and their community structure.

We used SOM to understand the characteristics of each individual fish collected. As shown in Figure 4a, the SOM result by each sampling gear, sampling location, and period did not show any significant pattern. In contrast, sampling gears showed a pattern that indicated a significant relationship with the standard length and biomass of fish. Among the sampling gears, the gill net showed the greatest similarity to the standard length pattern, whereas the fyke net showed the least similarity to the standard length pattern. These results indicate that the gill net collected larger fish than the other sampling gears, whereas the fyke net collected smaller fish. The cast net collected mostly medium-sized fish, thus explaining the lack of pattern similarity between this gear type and the standard length.

We confirmed that the standard length distribution of dominant species increased from clusters 1 to 4 (Figure 4c). Of the 2142 individuals of *L. macrochirus* analyzed for the standard length distribution, 1230 individuals were in cluster 1, and most of them were collected from the fyke net. In a study of age versus total length of *L. macrochirus* in Korea, individuals less than 90 mm in standard length were classified as one-year-old fish. Similarly, the standard length range of fish in cluster 1 was 22–59 mm (mean = $38.3 \pm 10.9$, $n = 1230$), indicating that they were all one-year-old individuals. The fyke net has been shown to catch fish that are generally smaller than fish caught by most other sampling gears [53,54]. Therefore, depending on the mesh size selected for each sampling gear, gill nets and fyke nets can be used in combination to sample different life stages of the same fish species.

Since the fyke net effectively samples a wide variety of fish species and large numbers of individuals, this one sampling gear alone can be used to approximate the community structure of the fish in each reservoir. However, the fyke net cannot be used to detect changes in the size of fish, cannot effectively collect fish that inhabit the littoral zone, and can lead to errors in determining the community structure of fish assemblages due to its high species selectivity. Therefore, to investigate a reservoir's overall fish community structure, it is important to select sampling gears that are well suited for the purpose of the investigation and the site-related characteristics.

## 5. Conclusions

Various studies on the comparison of sampling gear have been conducted with the development of sampling gear. However, many studies have investigated how many fish are collected in the sampling gear. In this study, fish were collected from three reservoirs selected by size using four sampling gears (kick net, cast net, gill net, and fyke net). As a result, a study was conducted regarding which factors affect the sampling gear through SOM analysis. In general, SOM is mainly used to analyze the effects of living organisms and environmental factors, but in this study, the relationship between the sampling gear and fish characteristics was visualized. This study provides the characteristics of the fish collected according to the sampling gear. The information on the characteristics between the sampling gears and standard length can be used for juveniles and resource management depending on the fish species. In addition, these results showed a patterned relationship between the type of sampling gear and the characteristics of the fish collected by each gear, suggesting that researchers need to select appropriate sampling gears based on their study objectives and the environmental characteristics of their sampling sites.

**Author Contributions:** Conceptualization, Y.-S.P., K.-H.H. and I.-S.K.; methodology, T.-S.Y., C.W.J., and I.-S.K.; formal analysis, T.-S.Y. and C.W.J.; investigation, T.-S.Y. and C.W.J.; writing—original draft preparation, T.-S.Y.; writing—review and editing, all authors.; visualization, T.-S.Y. and C.W.J. All authors have read and agreed to the published version of the manuscript.

**Funding:** This research was funded by the National Research Foundation of Korea (grant number NRF-2018R1A6A1A03024314) and the Korea Environment Industry and Technology Institute (KEITI) through the Aquatic Ecosystem Conservation Research Program funded by the Korea Ministry of Environment (MOE) (2020003050003).

**Institutional Review Board Statement:** Not applicable.

**Data Availability Statement:** Not applicable.

**Conflicts of Interest:** The authors declare no conflict of interest.

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
