# Peer review of "Characterization of Fish Assemblages and Standard Length Distributions among Different Sampling Gears Using an Artificial Neural Network"

_fishes, doi:10.3390/fishes7050275_

Round 1

Reviewer 1 Report

Paper ID: fishes-1917877-

Paper Title: Characterization of Fish Assemblages and Standard-Length Distributions Among Different Sampling Gears Using an Artificial Neural Network

Reviewer comments: The authors compare the community structure of fish assemblages sampled 16 using four sampling gears (kick net, cast net, gill net, and fyke net) in Singal (SG), Yedang (YD), and 17 Juam (JA) reservoirs and experimentally analyze characteristics of fishes collected by each sampling 18 gear. Results are presented in a meaningful way; the materials and methods are clearly stated. The topic is interesting, and it may be useful for the scientific and technical community. However, the authors need to make a major revision to improve the paper. Please see my comments below:

Concern 1. A complete review of grammar and style must be done in order to improve the quality of the publication.

Concern 2. The authors did not summarize the results in the abstract and also give the paper interest. I think the authors should rewrite the abstract.

Concern 3. The introduction section is poorly written. In the literature review of the introduction section, the authors just reported what have done the other authors in the past decades. The authors should highlight the weaknesses/research gaps of the previous relevant studies.

Concern 4. How the authors selected the area and the different fishing gears? What was the objectives to selected theses fishing gears?

Concern 5. The authors have to explain in detail the experimental process of the water quality.

Concern 6. The authors should add a flowchart to explain how the self-organizing map have been applied. Which software have been used?

Concern 7. The authors talk about the CPUE in the abstract, but I don't see where they analyzed. I think the authors should choose some main species common to each area to determine the CPUE for a better comparison. same with fishing gear.

Concern 8. In term of fish length and gear selectivity, what was the best fishing gear? I think that the authors have to add the selectivity graphics to better compare the catch performance of each gear.

 Author Response

Thank you for reviewing the paper.
It has been modified to reflect the reviewer's comments, and the modifications are written in the attached file.

Reviewer 2 Report

Generally I felt the abstract and introduction to this paper set out clearly the uncertainty in the use of different netting types in inshore/freshwater areas as they lead to different biases in size and shape of catch (and its effects of species selectivity) as well as the areas of the water body they can be deployed to. However, I feel there are large sections of the methods and results section either missing, unclear or vague with regard to the analysis, which undermines the conclusions.

Major Comments:

Firstly, comparisons of abiotic aspects of the different water bodies sampled are made yet no mention of this is within the methods, p values are quoted yet no statistical tests are mentioned throughout the manuscript.

Secondly, diversity and dominance metrics are quoted and mentioned yet no description of which metrics or their calculation is mentioned in the methods or results. Further, but less importantly mean values plus/minus are mentioned but the plus or minus is not stated. Is it Standard Error, 1 Standard Deviation, 2 Standard Deviations, 95% confidence, 87% confidence? Everyone has different conventions and preferences and most are arbitrary in my opinion but whichever is chosen needs to be stated.

Thirdly, (and i will mention i am not an expert or practitioner of Self Organising Maps) there seems to be the beginnings of a description of the method but many elements are missing, vague or misleading. For example, no mention of clustering was made in the methods. A mention of Bray-Curtis dissimilarity metric is made, yet a graph depicting a dendrograph (Figure 4a) displays Euclidean distance and heirarchical cluster is mentioned in the fig caption in relation to the graph, alongside U-Matrix values, yet their use and application is vague in the methods. A minor comment on the colour scale is that I assumed dark red was high and white was low, this is shown to be incorrect in the figure caption so fine as is but slightly confusing to me, especially without an in figure legend.

Finally, and potentially most importantly, I feel the analysis has been cut off short before the main result could be formally described. The SOM, from the authors description but unclear to this reader from the figures, has highlighted the difference between net types into clusters and that the main partition between clusters was driven by Standard Length. Once this information was discovered I would have expected a formal analysis of the standard lengths captured by the different net types, taking into account the fact of different species having different other traits/characteristics. This could be done in a multitude of ways (Multiple regression models/ANOVAS, but repeated p values need to be accounted for, larger more complex multilevel models (GLMMS) yet numbers of some species might create odd data distributions/biases in residual distributions so assumptions of the models used will need to be extensively considered).

Minor Comments: 

Line 30: Change "of" to "in"

Line 130-131: this should all be in the methods. 

Line 135-138: where do these p values come from? and where are the rest of any stats coefficients? 

Line 137: here and throughout plus/minus values given but not sure what they are.

Line 142: on line 131 WT is defined as Water Temperature yet this i assume is Weight?

Line 148: here and throughout the use of ~ is odd to me, normally i would interpret this symbol to mean "about/circa" yet here it is used to show a range which i was expect to see a -.

Line 166-171: these metrics are quoted but have not been introduced or defined.

Figure 2: where do these error bars come from? and how come they are only about the bars? To be honest I feel a bar plot is the incorrect display of this data, boxplots, mean points with error bars, cloud plots etc. would more accurately show the data.

Table 2: is the first table so should be 1, also odd horizontal line under first species entry. Relative Abundance is given as a % but again no calculation has been provided for it: relative to what? or is it the a percentage of the total abundance?

Figure 3a: I like the arrow showing the change in size of reservoir. However, I feel it is hard to see differences in standard length between the reservoirs. turning the graphs horizontal rather than vertical may aid the reader to distinguish differences in sizes across the reservoirs.

Figure 3b: same comment as above regarding inappropriate use of a barplot.

Line 185-186: I am not clear on why 7x10 matrix is used? 

Line 186-192: again this may be my ignorance on SOM but i dont see how the SOM is clustering data, isnt it the clustering of the dendrogram that shows that? Plus, how was the cut off for clustering decided upon? 

Line 194-195: "White circles represent high contributions while red circles represent low contributions" this is for the legend or methods not results section. As mentioned above i find this colour gradient counterintuitive, i normally expect simple colour schemes to show large values with darker colours/hues.

Lines 199-202: this "mid" value hasn't been defined, is it the difference of the range ie max(x)-((max(x)-min(x))/2) or is it a mean or median or mode?

Figure 4a and b: i find these figures unclear, especially what each hexagon/dot represents.

Figure 4c: This is a good figure, but the caption (Line 213-214) needs more explanation about what the numbers and abbreviations by each boxplot represent.

Line 253-257: this seems to say the fyke net can capture lots of different species but then says it is very selective? this makes no sense to me. This again is countered in line 285-286.

Line 294-303: is a nice over view of the study but i dont think it is a conclusion, it might be better at the beginning of the discussion.

Line 304: here and throughout you state the SOM as clustering the data but from my understanding it seems the SOM carried out some grouping of nodes then a clustering method was used. However, this may be my misinterpretation but if it is then I think more clear explanation is needed. If the SOM is doing the classifying, why is clustering by hierarchical clustering needed and by extension what is the use then of Bray Curtis/ Euclidean distance? 

Discussion: the discussion is fine, yet due to the short comings of the methods and results will need a rewrite once new results are provided. Especially the conclusions section, which currently focuses on the SOM and a "Strong patterned relationship between the type of sampling gear and the characteristics of fish", which I feel wasn't clear in any of the results section apart from in Figure 3c, which showed clear size differences across the clusters. Therefore, I feel the narrative of how those clusters were made and further analysis into the relationship between gear type and size structure of the fish is missing and would greatly improve this manuscript.

Author Response

Thank you for reviewing the paper.
It has been modified to reflect the reviewer's comments, and the modifications are written in the attached file.

Round 2

Reviewer 1 Report

The authors have made corrections based on the reviewer's suggestions,  the reviewer believes meet the requirements for publication.

Author Response

Thank you for your suggestion.

We conducted small revised to English expressions such as spelling.

Again, thank you for giving us the opportunity to strengthen our manuscript with your valuable comments and queries.

Reviewer 2 Report

The authors have managed to address many of my comments, but a few have not been addressed nor included in response to reviewers document.

As mentioned I believed that a formal analysis (glm or similar) to assess the differences in Standard Length across reservoirs as briefly displayed by figure 4c, could provide insight into the main driver (found from the previous) of variation between reservoirs. 

Author Response

Thank you for your suggestion.

But we do not have enough time to consider your suggestion during the minor review (only one day). We are sorry for not being able to reflect your request. In addition, small revised were made to English expressions such as spelling.

Again, thank you for giving us the opportunity to strengthen our manuscript with your valuable comments and queries.

Ln 56: multimesh-> multi-mesh

Ln 120: Fishbase-> FishBase

Ln 214: re servoirs-> reservoirs

Ln 266: Hypomesus olidus-> H. olidus